# Changes in the free-energy landscape of p38α MAP kinase through its canonical activation and binding events as studied by enhanced molecular dynamics simulations

Antonija Kuzmanic[1], Ludovico Sutto[2], Giorgio Saladino[2], Angel R Nebreda[1,3], Francesco L Gervasio[2]*, Modesto Orozco[1,4,5]*

[1]Institute for Research in Biomedicine (IRB Barcelona), The Barcelona Institute of Science and Technology, Barcelona, Spain; [2]Department of Chemistry, University College London, London, United Kingdom; [3]Catalan Institution for Research and Advanced Studies (ICREA), Barcelona, Spain; [4]Joint BSC-CRG-IRB Program in Computational Biology, Barcelona, Spain; [5]Department of Biochemistry, University of Barcelona, Barcelona, Spain

**Abstract** p38α is a Ser/Thr protein kinase involved in a variety of cellular processes and pathological conditions, which makes it a promising pharmacological target. Although the activity of the enzyme is highly regulated, its molecular mechanism of activation remains largely unexplained, even after decades of research. By using state-of-the-art molecular dynamics simulations, we decipher the key elements of the complex molecular mechanism refined by evolution to allow for a fine tuning of p38α kinase activity. Our study describes for the first time the molecular effects of different regulators of the enzymatic activity, and provides an integrative picture of the activation mechanism that explains the seemingly contradictory X-ray and NMR data.

*For correspondence: f.l. gervasio@ucl.ac.uk (FLG); modesto.orozco@irbbarcelona. org (MO)

**Competing interests:** The authors declare that no competing interests exist.

## Introduction

p38 Ser/Thr kinases are mitogen-activated protein kinases (MAPKs) involved in the regulation of multiple cellular processes, including cell proliferation, differentiation, senescence, and death (**Cuadrado and Nebreda, 2010**). The p38 subgroup of MAPKs comprises four members (α-δ) where only p38α is expressed ubiquitously at high levels (**Cuadrado and Nebreda, 2010**). p38 signaling is strongly activated by environmental stresses (e.g. osmotic shock, ionizing radiation), and biological stimuli, such as growth factors and inflammatory cytokines (**Trempolec et al., 2013**). Furthermore, p38α has been implicated in several pathological conditions, for example, chronic inflammatory diseases, cancer, and heart and neurodegenerative diseases (**Yong et al., 2009**, **Denise Martin et al., 2012**), which is why elucidation of its activation mechanism is of therapeutical importance (**Yong et al., 2009**; **Hammaker and Firestein, 2010**).

Like most kinases, p38α is composed of two lobes: the smaller N-terminal lobe, consisting mostly of β-sheets, and the α-helical C-terminal lobe. The lobes are linked by a flexible hinge that forms the ATP-binding site together with structural elements from both lobes (marked regions in **Figure 1**). In the canonical activation pathway, MAPK kinases (MAP2Ks) dually phosphorylate the TGY sequence in the activation loop (A-loop; **Figure 1a** and **Video 1**) which, according to X-ray crystallography (**Zhang et al., 2011a**), triggers its large conformational rearrangements and the formation of a

**Figure 1.** Structural models of (**a**) the inactive (PDB ID: 3S3I), and (**b**) the active, dually phosphorylated, p38α (PDB ID: 3PY3). Main structural elements are colored as follows: A-loop in purple (inactive) and green (active), α$_C$ helix in red, L16 loop in slate, P-loop in cyan, hinge in yellow, and MKI in orange. Key residues are shown as sticks and labeled appropriately. The ATP-binding site is indicated by the grey filled ellipse, while the α$_E$ helix, and CD and ED sites by the green dashed ellipse.

characteristic $\beta$-sheet motif away from the ATP- and substrate-binding sites (*Figure 1b*). The new position of the A-loop brings N- and C-lobes closer and facilitates the reorientation of key residues which participate in the stabilization of ATP and the catalytically crucial Mg$^{2+}$ ions (*Taylor and Kornev, 2011*) (i.e. the universally conserved K53, E71, and D168). Moreover, p38α and other MAPKs possess two distinct structural elements (*Figure 1*): (1) the MAPK insert (MKI), which forms a lipid-binding site (*Diskin et al., 2008*) and (2) the L16 loop that extends from the C-lobe to the N-lobe

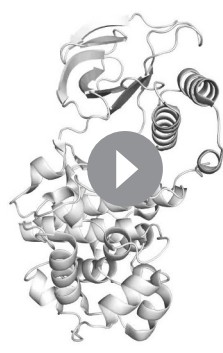

**Video 1.** Detailed presentation of p38α structural elements and binding sites which play an important role in its activation.

where it folds into the C-terminal L16 helix. At its C-lobe segment, the L16 loop contains an acidic patch, called the common docking (CD) motif (*Tanoue et al., 2000*), which together with the hydrophobic groove, formed by the linker between $\alpha_D$ and $\alpha_E$ helices, and the $\beta_7$-$\beta_8$ reverse turn (also termed the ED site) (*Chang et al., 2002*) (*Figure 1a*), defines the peptide docking site recognized by the conserved docking motif present in activators, substrates, and regulators of MAPKs. The docking site enhances the specificity of MAPKs' interactions, as well as its activity, albeit through an unknown molecular mechanism (*Tanoue et al., 2000*). All the structural elements are detailed in the *Video 1*.

The canonical activation mechanism outlined above, derived from the analysis of phosphorylated and unphosphorylated crystal structures of p38α (*Zhang et al., 2011a*) has recently been challenged by an NMR study of p38α in solution (*Tokunaga et al., 2014*), which surprisingly showed that the dual phosphorylation of the A-loop induces only small conformational changes to p38α, raising doubts about the reliability of conformational states associated with the canonical activation. To understand the changes that take place upon p38α phosphorylation and to explain the observed discrepancies between the NMR and X-ray results, we employed state-of-the-art simulation techniques. Specifically, we used unbiased molecular dynamics (MD) simulations and parallel tempering metadynamics (PT-metaD) (*Bussi et al., 2006*; *Bonomi and Parrinello, 2010*; *Sutto et al., 2012*), a method that combines an enhanced sampling technique with a multi-replica approach and is thus able to converge very complex conformational free-energy surfaces as a function of a few relevant variables, as has been shown previously for several protein kinases (*Berteotti et al., 2009*; *Lovera et al., 2012*; *Sutto and Gervasio, 2013*; *Lovera et al., 2015*; *Marino et al., 2015*) and other proteins (*Bonomi et al., 2007*; *Papaleo et al., 2014*; *Lambrughi et al., 2016*). In total, we obtained a high level of sampling (>115 µs) which allowed us to explore the molecular mechanism of p38α canonical activation in unprecedented detail and reconcile the apparently contradictory X-ray and NMR data.

## Results

We performed PT-metaD in the well-tempered ensemble to obtain free-energy surfaces (FESs) for human p38α in four different states: (1) the unphosphorylated, (2) the apo dually phosphorylated, (3) the dually phosphorylated state with bound ATP-Mg$^{2+}$, and (4) the dually phosphorylated state with bound ATP-Mg$^{2+}$ and MK2 docking peptide (*Figure 2* and *Figure 2—figure supplement 1*), obtaining total sampling times of 24.25, 24.85, 10.35, and 9.86 µs, respectively. To analyze the systems, we projected the FESs along two collective variables (CVs) - CV$_1$ and CV$_2$, both of which are based on the A-loop contacts that measure the distance from the crystallographically observed inactive (CV$_1$) and active (CV$_2$) conformations (see Materials and methods), and which were successfully used previously to study protein kinases (*Sutto et al., 2012*; *Sutto and Gervasio, 2013*; *Marino et al., 2015*). When CV$_1$ < 0.2 and CV$_2$ > 0.8, then the system assumes an inactive conformation (as in *Figure 1a*). On the other hand, when CV$_1$ > 0.6 and CV$_2$ < 0.2, the A-loop has formed the characteristic $\beta$-sheet motif (as in *Figure 1b*) and the enzyme has adopted the canonical active state. As the representative structure of each highlighted minimum, we chose the central structure of the most populated cluster present in the minimum (see Materials and methods for further details).

### Unphosphorylated p38α

FES of the unphosphorylated state (upper left panel, *Figure 2*) and the representative structures of the explored minima (*Figure 2* and *Figure 2—figure supplement 1*, enlarged structures in *Figure 2—figure supplement 2*) show that the unphosphorylated protein neither explores the active

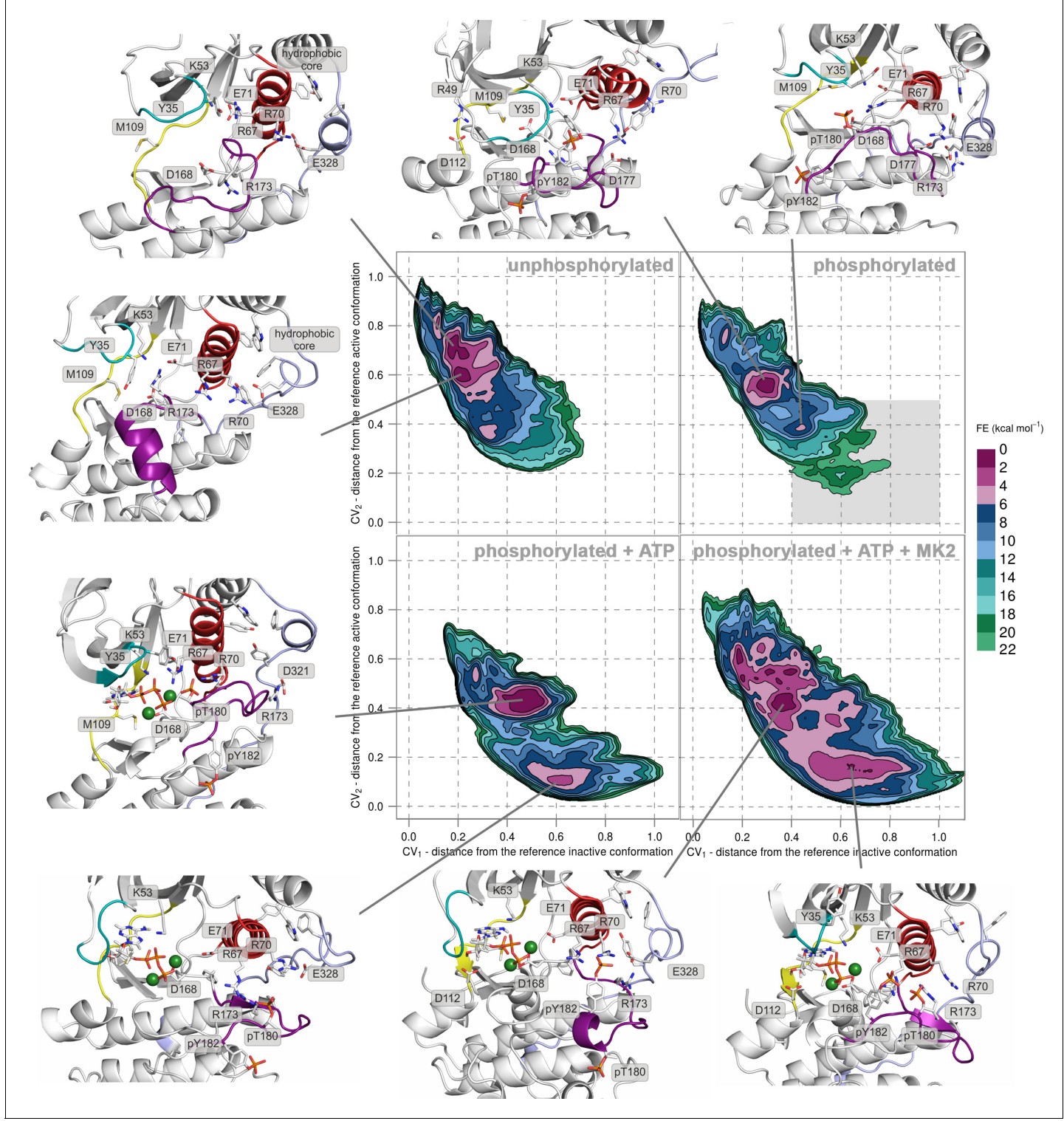

**Figure 2.** Free-energy surface for p38α (apo unphosphorylated, apo dually phosphorylated, dually phosphorylated with bound ATP-Mg$^{2+}$, and dually phosphorylated with bound ATP-Mg$^{2+}$ and MK2 peptide) as a function of CV$_1$ and CV$_2$ which indicate the distance from the reference inactive and active structures, respectively. The contour lines are drawn every two kcal mol$^{-1}$. Representative structures for the selected minima are shown and colored as follows: A-loop in purple, α$_C$ helix in red, L16 loop in slate, P-loop in cyan, and hinge in yellow. Key residues are shown as sticks and labeled appropriately. Part of the FES of the phosphorylated p38α that is populated by structures which are likely to bind ATP are highlighted with a grey rectangle (upper right panel).

*Figure 2 continued on next page*

*Figure 2 continued*

The following figure supplements are available for figure 2:

**Figure supplement 1.** Free-energy surface for p38α (apo unphosphorylated, apo dually phosphorylated, and dually phosphorylated with bound ATP-Mg$^{2+}$, and dually phosphorylated with bound ATP-Mg$^{2+}$ and MK2 peptide) as a function of CV$_1$ and CV$_2$ which indicate the distance from the reference inactive and active structures, respectively.

**Figure supplement 2.** Representative structures of selected free-energy minima calculated for p38α in the apo unphosphorylated state located at: (a) (CV$_1$, CV$_2$) = (0.2, 0.7), (**b**) (CV$_1$, CV$_2$) = (0.2, 0.6), (**c**) (CV$_1$, CV$_2$) = (0.1, 0.8), and d) (CV$_1$, CV$_2$) = (0.3, 0.35).

**Figure supplement 3.** Representative structures of selected free-energy minima calculated for p38α in the apo phosphorylated state located at: (a) (CV$_1$, CV$_2$) = (0.3, 0.55), (**b**) (CV$_1$, CV$_2$) = (0.45, 0.4), (**c**) (CV$_1$, CV$_2$) = (0.15, 0.75), and d) (CV$_1$, CV$_2$) = (0.6, 0.2).

**Figure supplement 4.** Histogram of the minimal distances between residues R49 and D112 calculated for the structures of the most populated cluster in the global minimum of the dually phosphorylated p38α.

**Figure supplement 5.** Representative structures of selected free-energy minima calculated for p38α in the phosphorylated state with bound ATP-Mg$^{2+}$ located at: (a) (CV$_1$, CV$_2$) = (0.5, 0.4), (**b**) (CV$_1$, CV$_2$) = (0.6, 0.1), and (**c**) (CV$_1$, CV$_2$) = (0.25, 0.5).

**Figure supplement 6.** Representative structures of selected free-energy minima calculated for p38α in the phosphorylated state with bound ATP-Mg$^{2+}$ and MK2 peptide located at: (a) (CV$_1$, CV$_2$) = (0.7, 0.1), and (b) (CV$_1$, CV$_2$) = (0.4, 0.4).

conformation, nor does it populate a state able to accommodate ATP. Specifically, in the more inactive of the two free-energy minima - (CV$_1$, CV$_2$) = (0.2, 0.7) (upper left panel in *Figure 2*, *Figure 2—figure supplement 2a*), the A-loop is positioned over the C-lobe forming contacts with residues involved in the substrate binding and phosphotransfer (*Table 1*). The P-loop is quite flexible and the K53-E71 salt bridge, which stabilizes ATP, is displaced from the C-lobe due to the broken regulatory spine that is otherwise formed with D168 upon ATP binding (*Taylor and Kornev, 2011*). Furthermore, E71 interacts with the MAPK-conserved R67 from the α$_C$ helix that stabilizes pT180 in the active form (*Figure 1b*, present in >90% of structures in the most populated cluster). On the other hand, the A-loop of the second minimum contains a short helix which places the conserved R173 in direct contact with the K53-E71 salt bridge, as well as D168 of the conserved DFG motif (*Figure 2*, *Figure 2—figure supplement 2b*), while the P-loop is partly collapsed into the ATP-binding site. We observe similar autoinhibitory interactions for the other two higher energy minima explored by p38α (*Supplementary file 1*, *Figure 2—figure supplement 2c,d*). In both cases, R173 is in contact with D168 which otherwise stabilizes ATP upon binding (*Kumar et al., 1995*; *Bellon et al., 1999*).

**Table 1.** Interactions of key residues observed in the free-energy minima. Residue pairs with interaction occupancy >75% in the most populated clusters of the selected minima are bolded, while the occupancy for the rest of the pairs is in the range of 60–75%.

| System | Minimum | Residue pairs |
|---|---|---|
| **p38α** | 0–2 kcal mol$^{-1}$ (CV$_1$, CV$_2$ = 0.2, 0.7) | D176-R189, **D177-R186**, E178-K152, E178-R173 |
| | 0–2 kcal mol$^{-1}$ (CV$_1$, CV$_2$ = 0.2, 0.6) | **R173-E71**, D168-R173, D177-R186 |
| **p38α-pTpY** | 0–2 kcal mol$^{-1}$ | E178-R149, E178-R189, D177-R67, pY182-R173 |
| | 4–6 kcal mol$^{-1}$ (CV$_1$, CV$_2$ = 0.45, 0.4) | D176-R149, D176-R189, D177-R70, pT180-R67, R173-E328 |
| **p38α-pTpY (ATP-Mg$^{2+}$)** | 0–2 kcal mol$^{-1}$ | **pT180-R67**, pT180-R70, **pY182-R186**, pY182-R189, E178-R70 |
| **p38α-pTpY (ATP-Mg$^{2+}$ + MK2 peptide)** | 0–2 kcal mol$^{-1}$ (CV$_1$, CV$_2$ = 0.7, 0.1) | **pT180-R67**, pT180-R173, D112-ATP |
| | 0–2 kcal mol$^{-1}$ (CV$_1$, CV$_2$ = 0.4, 0.4) | pY182-R67, pY182-R70, D177-R173, D112-ATP |

Furthermore, in the more inactive minimum (*Figure 2—figure supplement 2c*), the P-loop is collapsed into the ATP-binding site; while the other explores a conformation reminiscent of the Src-like inactive conformation (*Xu et al., 1999*) (*Figure 2—figure supplement 2d*) in which the A-loop forms a short helix stacked against the $\alpha_C$ helix. This configuration has been previously observed also in CDK (*De Bondt et al., 1993*) and EGFR kinases (*Sutto and Gervasio, 2013*). These findings are in perfect agreement with experimental studies reporting that the unphosphorylated p38α shows almost no affinity for ATP (*Tokunaga et al., 2014*; *Frantz et al., 1998*), confirming the strong regulation of the enzyme by phosphorylation.

To confirm that the extended conformation of the A-loop attributed to the active state is highly unstable in the unphosphorylated protein, we used unbiased MD simulations that were started from the conformation observed in the X-ray structure of p38α-pTpY (PDB ID: 3PY3) (*Figure 1b*) (see Materials and methods). We performed ten 500-ns-long independent runs at 380 K, all of which quickly showed a change in the A-loop conformation toward the inactive state and sampled conformations that fall in the regions explored in metadynamics simulations. Furthermore, in one of the independent runs the A-loop managed to break the β-sheet motif in the simulated time, so we extended this run to 2.675 μs to allow the protein to transition to the inactive state (*Figure 3a*). Indeed, the protein mostly remained in the global minima observed in metadynamics simulations, and the final conformation showed very similar features to the representative inactive structures (*Figure 3b*). Root-mean-square fluctuations (RMSF) reveal a high flexibility of several structural elements besides the A-loop (i.e. $\alpha_C$, $\alpha_D$, and the MKI helices, *Figure 3b*) that occurs due to the opening of the lobes and the local unfolding (in the case of $\alpha_D$ helix). Both these events have been previously linked to large conformational changes occurring in other kinases (*Sutto and Gervasio, 2013*; *Whitford et al., 2007*; *Shan et al., 2013*) but have been so far unobserved for p38α.

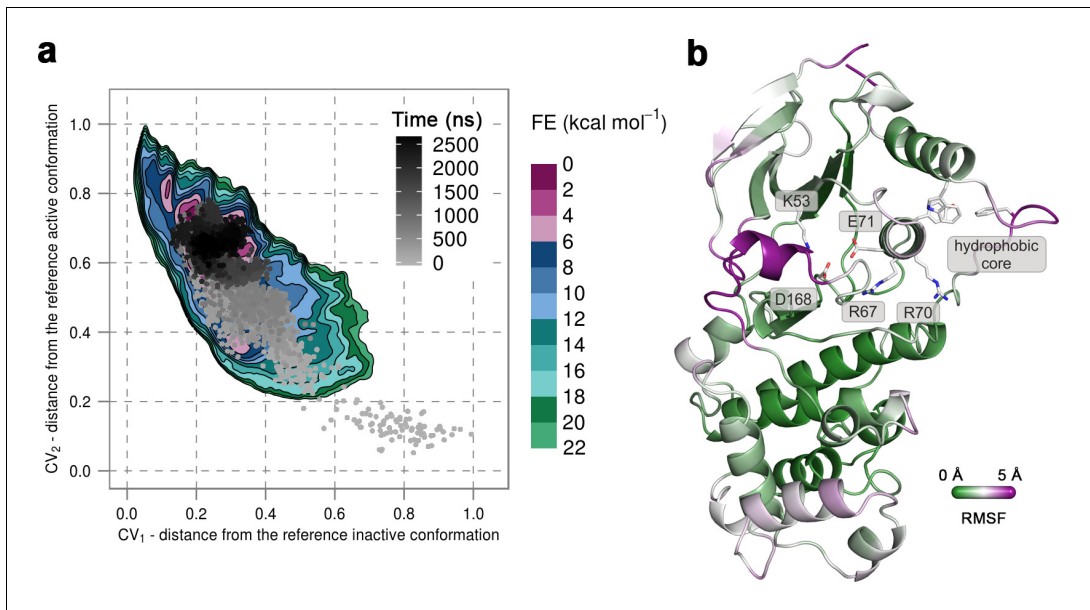

**Figure 3.** (a) Free-energy surface for the unphosphorylated p38α, obtained from metadynamics simulations, as a function of $CV_1$ and $CV_2$ which indicate the distance from the reference inactive and active structures, respectively. The contour lines are drawn every two kcal mol$^{-1}$. The CV values for the unbiased MD simulation of the same system are drawn every 0.5 ns and shown in a grey gradient as a function of time. (b) The conformation assumed by the protein at the end of the unbiased MD simulation. The backbone is colored according to the RMSF values, with purple indicating higher flexibility regions, while the key residues are shown as sticks and labeled appropriately.

## Dually phosphorylated p38α

Surprisingly, once p38α is dually phosphorylated, it explores conformations that are not largely different from the ones observed for the unphosphorylated protein (upper right panel in *Figure 2* and *Figure 2—figure supplement 1*, enlarged structures in *Figure 2—figure supplement 3*), contrary to the reported X-ray data (*Zhang et al., 2011a*). In the global minimum, the A-loop moves away from the ATP-binding site, mostly due to the electrostatic interactions formed by the phosphorylated residues which subsequently also stabilize the P-loop and prevent it from collapsing into the ATP binding pocket (*Table 1*). However, the sampled conformation is unlikely to bind ATP due to the formation of a salt bridge between R49 (positioned beneath the P-loop) and D112 (found at the beginning of the $α_D$ helix) (*Figure 2*, *Figure 2—figure supplement 3a*; distance histogram is shown in *Figure 2—figure supplement 4*). This interaction, which seems to be specific for the phosphorylated protein, locks the hinge and occludes the ATP-binding site. However, the protein also explores a higher energy minimum that comes closer to the active one (*Figure 2*, *Figure 2—figure supplement 3b*): the A-loop extends toward the L16 loop, yet does not form the β-sheet motif associated with the active conformation. Regardless, the higher energy conformation allows the A-loop to form a number of electrostatic interactions with the $α_C$ helix and the L16 loop (*Table 1*), while pT180 forms a salt bridge with the highly conserved R67 preventing the P-loop from collapsing into the ATP-binding site. Furthermore, the $α_D$ helix partly unfolds allowing the ATP-binding pocket to assume a more open conformation with the key residues aligned for ATP binding. This observation explains the protein's low affinity toward ATP in the phosphorylated state (*Tokunaga et al., 2014*; *Frantz et al., 1998*) and demonstrates that the dual phosphorylation can help define a receptive ATP-binding site but is not sufficient for the protein to adopt the conformation captured by X-ray crystallography (*Figure 1b*). In fact, to populate that state, we would need to add 16–18 kcal mol$^{-1}$ of free energy to the phosphorylated system (minimum located at ($CV_1$, $CV_2$) = (0.6, 0.2) in the upper right panel of *Figure 2*, *Figure 2—figure supplement 3d*).

To further confirm the instability of the conformation observed by X-ray crystallography, we again used unbiased MD simulations that were started from the said conformation (PDB ID: 3PY3) (*Figure 1b*) (see Materials and methods). We performed ten 1-μs-long independent runs at 380 K, most of which showed a change in the A-loop conformation toward the regions explored by the protein in metadynamics simulations (upper right panel in *Figure 2*). Furthermore, in one of the independent runs the A-loop broke the β-sheet motif, so we extended this run to 2 μs during which the protein settled in the aforementioned higher energy minima that seems likely to bind ATP (*Figure 4a*). The final conformation shows similar features to the representative structure of this minimum, that is, the A-loop is positioned away from the ATP-binding site and forms several electrostatic interactions with the $α_c$ helix and the L16 loop (*Figure 4b*). RMSF values reveal a high flexibility of several structural elements besides the loops (i.e. $α_D$, and the MKI helices, *Figure 4b*) that occurs due to local unfolding.

## Dually phosphorylated p38α with bound ATP-Mg$^{2+}$

The FES plot for the phosphorylated system with bound ATP-Mg$^{2+}$ complex shows a clear shift toward a more active conformation (i.e. a displacement toward right and bottom in the lower left panel of *Figure 2* and *Figure 2—figure supplement 1*, enlarged structures in *Figure 2—figure supplement 5*). While the representative structure of the global minimum does not form the β-sheet motif associated with the active state (*Figure 2—figure supplement 5a*), its A-loop extends toward the L16 loop and connects to the $α_C$ helix via the coordination of pT180 by R67 and R70 (as observed in the active X-ray structure in *Figure 1b*). The β-sheet motif does form in the higher energy minimum centered at ($CV_1$, $CV_2$) = (0.6, 0.1); however, its representative structure (lower left panel in *Figure 2*, *Figure 2—figure supplement 5b*) shows pT180 forming less contacts with the $α_C$ helix and the L16 loop (*Supplementary file 1*). The lack of these interactions, together with the disorder present in the P-loop, breaks the contact of E71 with Mg$^{2+}$ ions resulting in a less stable complex with ATP. Overall, albeit the binding of ATP stabilizes an active-like structure, it is not enough for the protein to display the active conformation observed in the crystal structure (*Figure 1b*). While the A-loop does adopt an extended conformation and forms the β-sheet motif at the higher energy state, it still remains quite mobile and the phosphorylated residues do not necessarily form all the contacts observed in the X-ray structures.

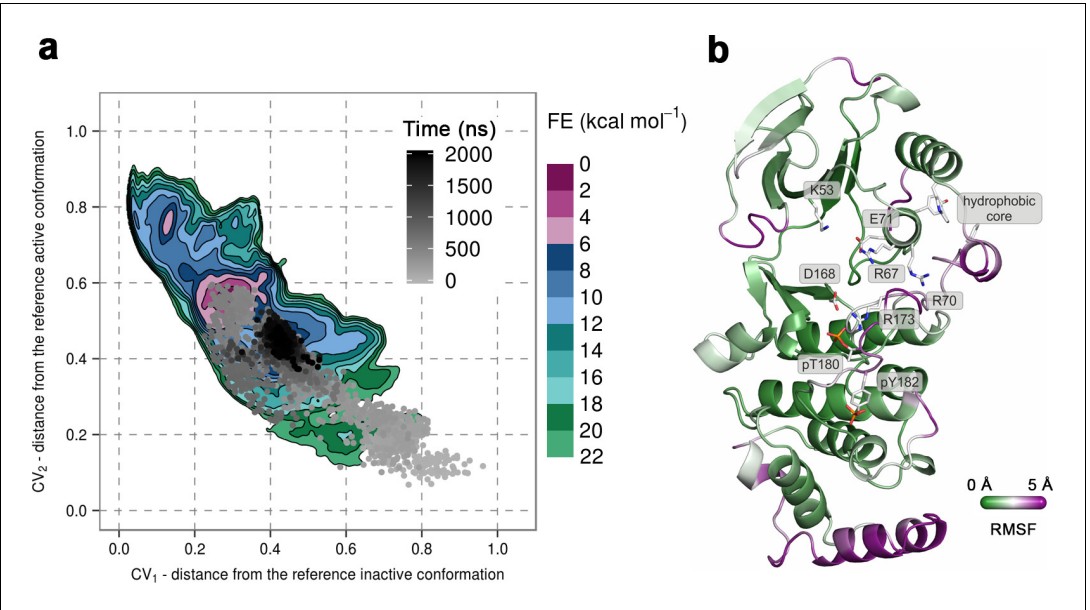

**Figure 4.** (a) Free-energy surface for the dually phosphorylated p38α, obtained from metadynamics simulations, as a function of $CV_1$ and $CV_2$ which indicate the distance from the reference inactive and active structures, respectively. The contour lines are drawn every two kcal $mol^{-1}$. The CV values for the unbiased MD simulation of the same system are drawn every 0.5 ns and shown in a grey gradient as a function of time. (b) The conformation assumed by the protein at the end of the unbiased MD simulation. The backbone is colored according to the RMSF values, with purple indicating higher flexibility regions, while the key residues are shown as sticks and labeled appropriately.

Once again, we used unbiased MD simulations to give further support to our free-energy surfaces. We performed ten 1-μs-long independent runs at 380 K of the dually phosphorylated p38α with bound ATP-$Mg^{2+}$ starting with the A-loop in the extended conformation (as in *Figure 1b*) (see Materials and methods). Most of the simulations explored only the local minimum close to the starting structure, while one of the runs showed more conformational diversity, so we extended it to 1.55 μs which allowed the protein to transition to the global minimum (*Figure 5a*). The final conformation shows the disruption of the β-sheet motif in the A-loop and the calculated RMSFs reveal its high flexibility (*Figure 5b*). Furthermore, pT180 remains in contact with the conserved arginine residues of the $α_C$ helix, while pY182 moves away from H228, similarly to the representative structure of the global minimum in the free-energy surface. Overall, RMSF values show a lower flexibility of the protein upon the addition of the ATP compared to the simulations of the apo systems (*Figures 3b* and *4b*).

## The active conformation

While the dual phosphorylation and the ATP-binding drive the enzyme toward an active conformation, there is an obvious need for additional elements to complete the transition. Particularly, we wondered whether the binding of the MK2 docking peptide could provide the necessary stability to form the β-sheet motif. The FES for this system indeed shows a deepening of the minimum at ($CV_1$, $CV_2$) = (0.7, 0.1) which is associated with the formed β-sheet motif (lower right panel in *Figure 2*, enlarged structures in *Figure 2—figure supplement 6*). The representative structure of the minimum (*Figure 2—figure supplement 6a*) confirms this observation and shows pT180 near the three conserved arginines (R67, R70, and R173), while pY182 again displays higher mobility and moves away from H228 (*Table 1*, *Figure 1b*). Furthermore, the addition of the MK2 peptide does not eliminate the global minimum at ($CV_1$, $CV_2$) = (0.4, 0.4), observed also in the presence of ATP (two lower panels in *Figure 2*). However, it flattens the FES which would indicate an easier transition between the two described minima. The representative structure of the minimum (*Figure 2—figure supplement*

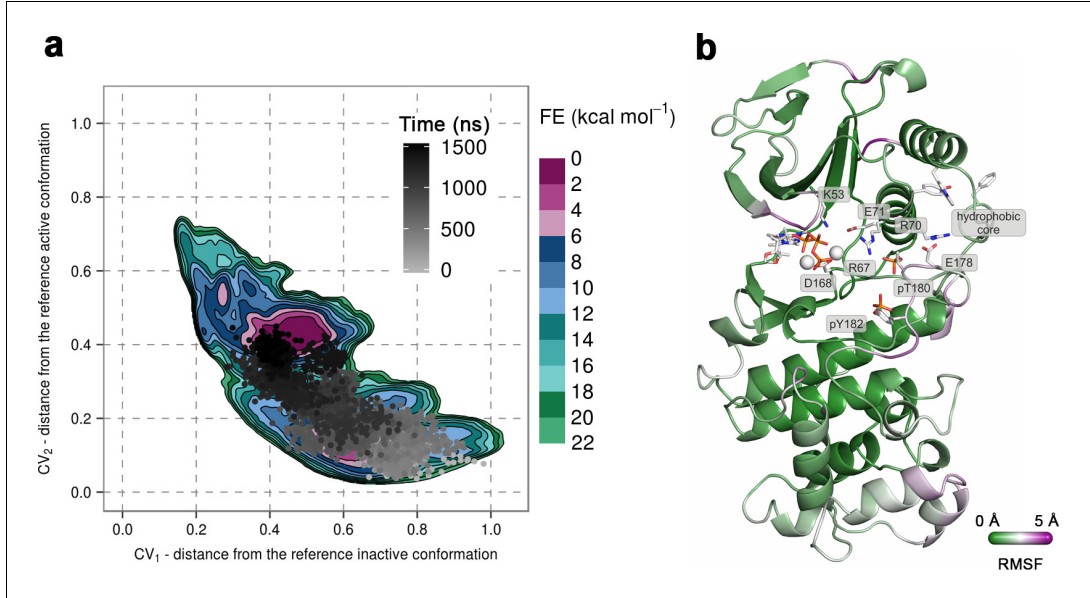

**Figure 5.** (a) Free-energy surface for the dually phosphorylated p38α with bound ATP-Mg$^{2+}$, obtained from metadynamics simulations, as a function of CV$_1$ and CV$_2$ which indicate the distance from the reference inactive and active structures, respectively. The contour lines are drawn every two kcal mol$^{-1}$. The CV values for the unbiased MD simulation of the same system are drawn every 0.5 ns and shown in a grey gradient as a function of time. (b) The conformation assumed by the protein at the end of the unbiased MD simulation. The backbone is colored according to the RMSF values, with purple indicating higher flexibility regions, while the key residues are shown as sticks and labeled appropriately.

*6b*) shows a conformation in which the A-loop is extended away from the ATP-binding site, yet without the formed β-sheet motif and with switched positions of phosphorylated residues compared to the X-ray structure (*Table 1*, *Figure 1b*).

In addition, we performed unbiased MD simulations of the dually phosphorylated p38α with bound ATP-Mg$^{2+}$ and the MK2 docking peptide. We generated ten 1-μs-long independent runs at 380 K, obtaining a total accumulated sampling time of 10 μs (see Materials and methods). The protein remains mostly in the regions associated with the extended conformation of the A-loop with formed β-sheet motif, but it also explores the conformational space closer to the other global minimum (also the global minimum in the system without the MK2 peptide) (*Figure 6a*). Interestingly, the N-terminal part of the MK2 docking peptide binds the ED site and stabilizes the α$_D$-α$_E$ linker hindering the flexibility of the α$_D$ helix and allowing the D112 to reorient and stabilize ATP by forming hydrogen bonds with ribose (*Figure 2—figure supplement 6*, *Figure 6b*). While we also observe this interaction in the absence of the MK2 peptide, it is not as stable as in its presence which explains the phosphotransfer enhancement upon MK2 peptide docking (*Tokunaga et al., 2014*).

Our results suggest that p38α predominantly samples conformations with the fully extended A-loop only upon double phosphorylation and the binding of ATP and the docking peptide. This might seem in contrast to the model obtained from X-ray crystallography (*Figure 1b*). However, careful analysis of the crystal contacts (which form between the molecules due to their orientation and packing within the crystal, *Figure 7*) reveals numerous interactions formed between functionally important structural elements, such as the L16 loop. These contacts mimic the stabilizing effects of p38α binders. Particularly, we found an atypically long expression His-tag of a neighboring molecule bound to the hydrophobic docking groove which, upon closer inspection (see below), has the same effect on the protein as the MK2 docking peptide did in our simulations.

To prove that this is the case, we simulated a crystal supercell of the apo dually phosphorylated p38α composed of 4 unit cells and 16 protein molecules (see Materials and methods), which allowed us to maintain crystal contacts and obtain 8 μs of sampling for ~170,000 atoms. As shown by *Figure 8*, the A-loop is very stable in the conformation captured by the X-ray model - the β-sheet motif

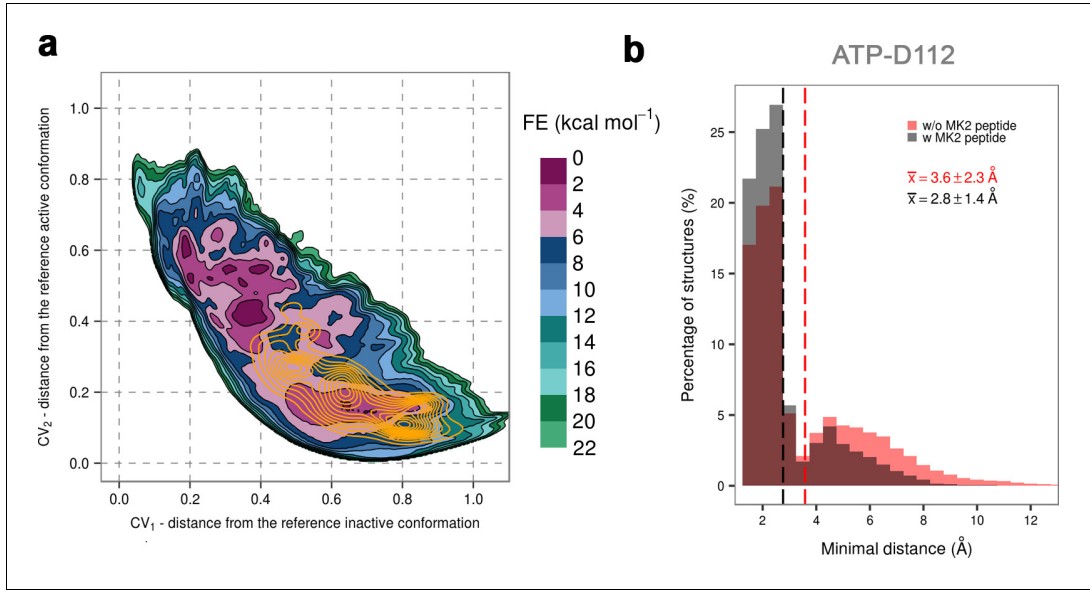

**Figure 6.** (a) Free-energy surface for the dually phosphorylated p38α with bound ATP-Mg$^{2+}$ and MK2 docking peptide, obtained from metadynamics simulations, as a function of CV$_1$ and CV$_2$ which indicate the distance from the reference inactive and active structures, respectively. The contour lines are drawn every two kcal mol$^{-1}$. The CV values (calculated every 0.5 ns) for the unbiased MD simulations of the same system are shown as orange contours based on their density. (b) Histograms of the minimal distances between ATP and D112 calculated from the unbiased simulations of the dually phosphorylated p38α with bound ATP-Mg$^{2+}$ with or without the MK2 docking peptide (colored grey and red, respectively). The average values are shown as dashed lines, while the bin size is 0.5 Å.

is unperturbed during the simulation, as well as the contacts made by pT180 with the three conserved arginines (R67, R70, and R173). However, pY182 is more mobile and at times breaks away from H228 and the conserved arginines of the P+1 loop (R186 and R189). This is consistent with the electron density which is not so well defined for pY182 as it is for the rest of the A-loop. Furthermore, we could also qualitatively match the B-factors reported for the X-ray structure (PDB ID: 3PY3) (*Figure 8b,d*), while the most mobile regions in our simulations (mainly, the N- and C-terminal tails) are missing from the deposited electron density. We would like to stress that it is impossible for our MD simulations to quantitatively match the reported B-factors for the following reasons: (1) While the crystals were grown at 293 K, the data were collected at 100 K. Since MD simulations and their algorithms have not been developed for such low temperatures, we performed our crystal simulations at 293 K which increased the overall atomic motions; (2) The experimental data collected for the crystals in general are time and ensemble averages of 10$^{23}$ molecules which are in the end represented by a single model, while its B-factors then reflect both the static and the dynamic disorder of the crystal. Finally, two independent studies had shown that the refinement process underestimates the disorder present in the crystal thereby producing lower B-factors (*Janowski et al., 2013*; *Kuzmanic et al., 2014*).

Thus, our analysis and simulations strongly suggest that the deposited crystal structure does not fully capture the apo conformation that p38α assumes under physiological conditions and it most likely suffers from artifacts due to the crystal contacts.

## Discussion

In the clever paraphrase of Tolstoy's Anna Karenina, Noble *et al.* (*Noble et al., 2004*) stated how all active kinases are alike but an inactive kinase is inactive after its own fashion - referring to the attractiveness of the inactive form in the everlasting search for inhibitor selectivity. Kinases, however, seem to have a few surprises left within their folds, even when it comes to the seemingly uniform

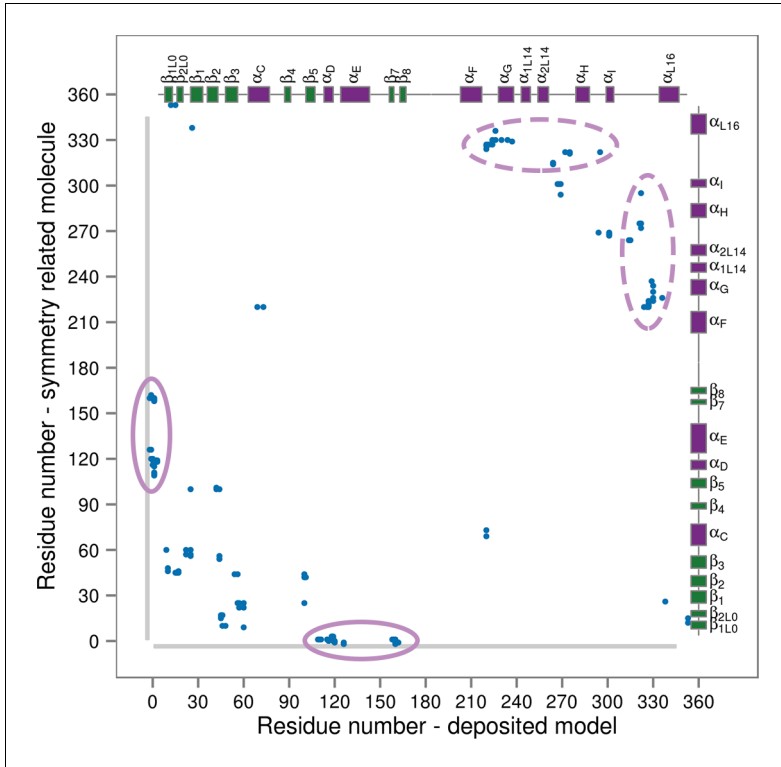

**Figure 7.** Contact map for interactions formed between the residues of the reported X-ray structure (PDB ID: 3PY3, p38α-pTpY) (x-axis) and its symmetry related neighbors (y-axis). The residues are considered to be in contact if the minimal distance between any of their atoms is under 4 Å. The contacts formed by the docked elements are indicated by the solid ellipses, while the ones formed between functionally important elements are indicated by the dashed ellipses. Secondary structure elements are also shown with α-helices in purple, β-sheets in green, and loops in grey. Missing residues are shown as light grey rectangles.

active state. In the case of p38α, the protein requires dual phosphorylation of the TGY motif for its activation and subsequent ATP binding. Based on the apo X-ray structure of p38α-pTpY (*Zhang et al., 2011b*), the phosphorylation event has been perceived as the trigger for the conformational change in which the A-loop moves away from the ATP-binding site enabling the N- and C-lobes to reorient and align the residues necessary for the ATP stabilization. Unlike CDK5 (*Berteotti et al., 2009*), EGFR (*Sutto and Gervasio, 2013*), and Src (*Shukla et al., 2014*; *Foda et al., 2015*), p38α never explores the αC-out conformation (neither in any of the deposited X-ray structures - *Figure 9a* nor in our simulations), in which the helix rotates by >45° (breaking the K53-E71 salt bridge in the process). This likely occurs due to the extensive range of contacts formed between the αC and the L16 helix, which is unique to MAPKs, indicating that the observed feature could be family-specific. Furthermore, MAP kinases contain two conserved arginine residues (R67 and R70) in the αC helix that readily form salt bridges with numerous negatively charged residues in the A-loop (*Figure 10*), thereby stabilizing various A-loop conformations which are not necessarily observed by X-ray crystallography. Despite an overwhelming amount of structural data (with 206 structures of human p38α in the PDB), just 47 structures have complete A-loops (i.e. without any missing or zero-occupancy residues), out of which only one structure contains the A-loop devoid of any crystal contacts (PDB ID: 2BAJ). Unfortunately, this structure has been solved in the DFG-out conformation with a bound inhibitor and it cannot be representative of the apo unphosphorylated state. However, the mere lack of information indicates just how flexible the A-loop is and why it is almost impossible to resolve it unless it forms crystal contacts. Our extensive simulations also indicate that the unphosphorylated p38α never explores a conformation that could possibly bind and stabilize ATP - mostly due to A-loop interactions with the DFG-Asp which is crucial for ATP

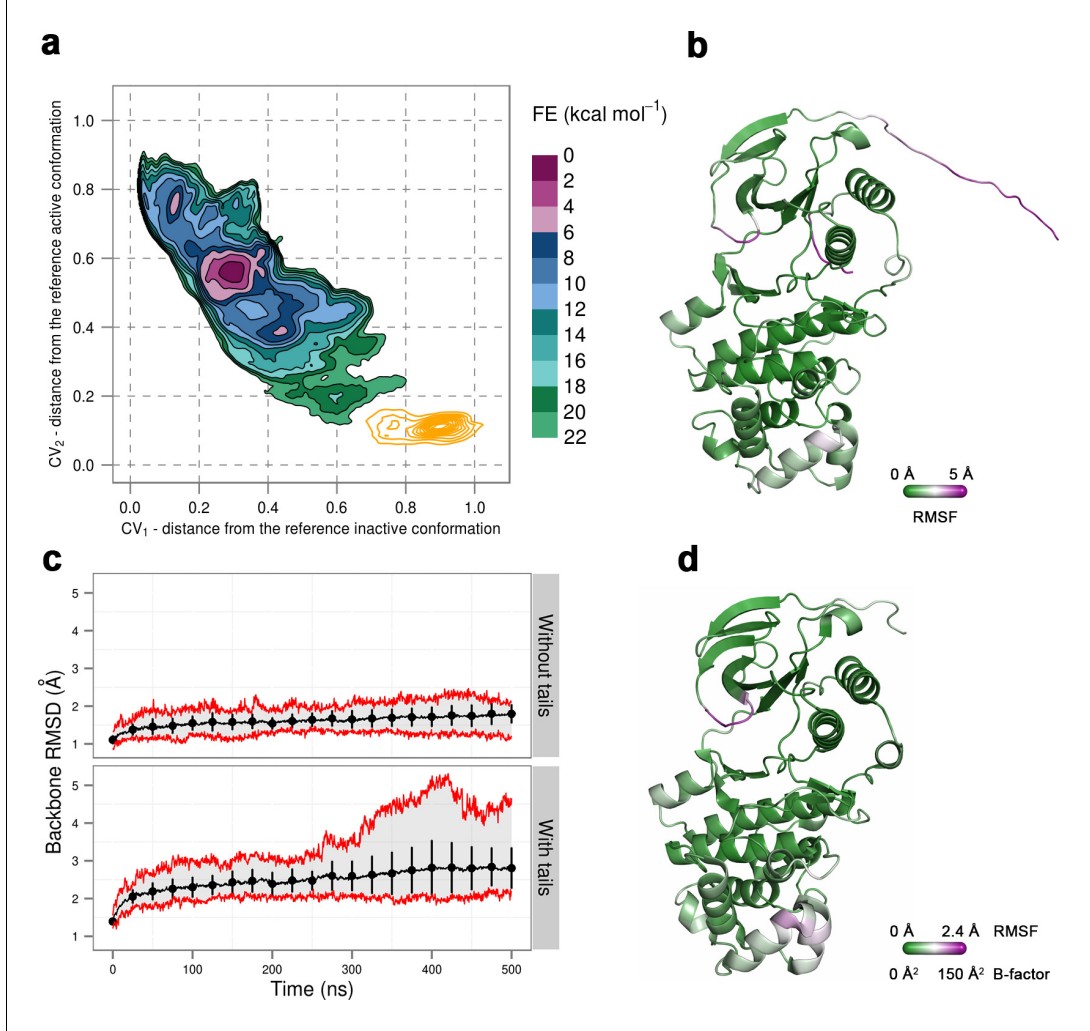

**Figure 8.** (a) Free-energy surface for the dually phosphorylated p38α, obtained from metadynamics simulations, as a function of $CV_1$ and $CV_2$ which indicate the distance from the reference inactive and active structures, respectively. The contour lines are drawn every two kcal mol$^{-1}$. The CV values (calculated every 0.5 ns) for the unbiased MD simulations of the 3PY3 crystal are shown as orange contours based on their density. (b) The average conformation of the protein calculated from the unbiased MD simulations. The protein is colored according to the calculated RMSF values after the backbone alignment with respect to the 3PY3 X-ray structure. (c) Backbone RMSD values calculated with respect to the 3PY3 X-ray structure either with or without N- and C-terminal tails. The average values over 16 molecules are shown by the black lines, the extreme values by the red lines, while the error bars depicting one standard deviation are shown every 25 ns. (d) X-ray structure of the dually phosphorylated p38α (PDB ID: 3PY3) colored by refined B-factors.

stabilization (*Kumar et al., 1995*). It is worth noting that related proteins, like ERK2, are able to bind ATP in the unphosphorylated state (*Lee et al., 2005*; *Sours et al., 2008*; *Canagarajah et al., 1997*; *Khokhlatchev et al., 1998*) most likely due to longer A-loops (*Figure 10*), which are then easier to stabilize away from the ATP-binding site.

Phosphorylation, however, does change the free-energy surface and enables the A-loop to form salt bridge interactions with the autoinhibitory arginine residues - R67 and R173. The former of the two, as mentioned above, is MAPK-specific and its importance is highlighted further by the R67A mutation that decreases p38α kinase activity (*Gum et al., 1998*), and the R67I somatic mutation that was found in colon adenocarcinoma (COSMIC: COSM1444062). The newly formed interactions keep the A-loop away from the ATP-binding site which is occluded by the formation of the R49-D112 salt

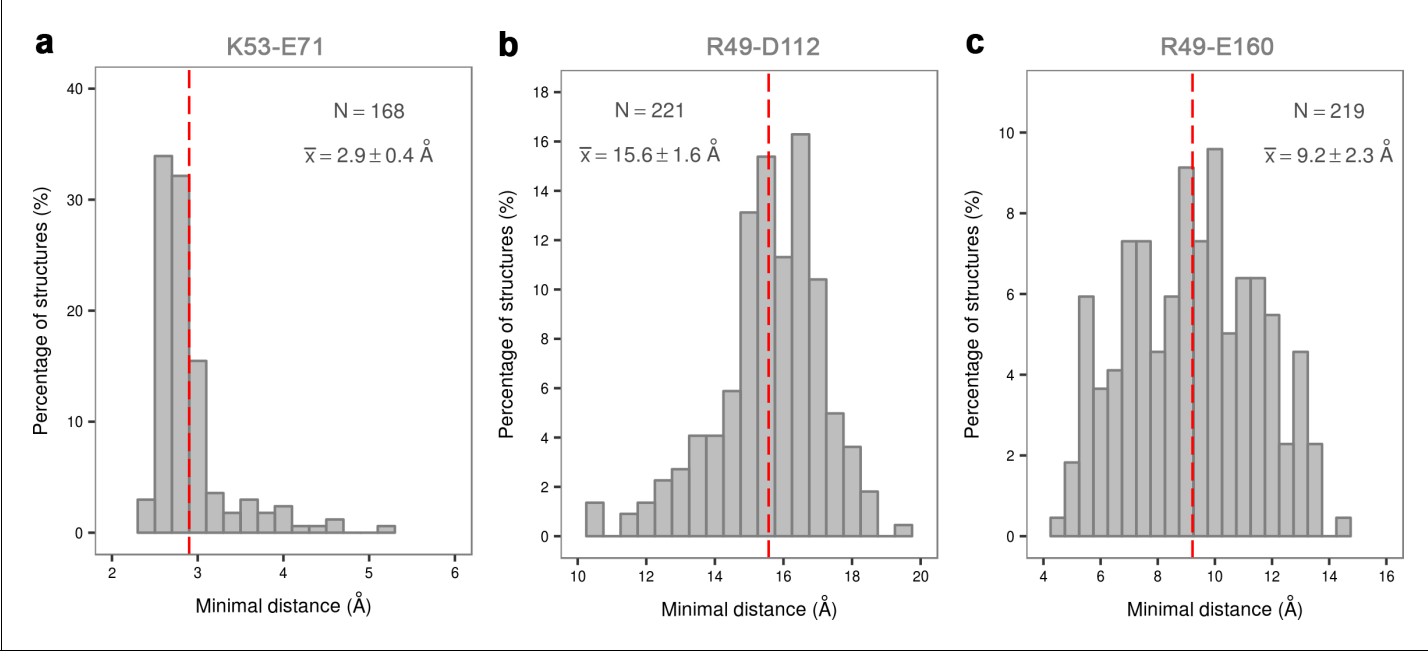

**Figure 9.** Histograms of the minimal distances between (a) K53-E71, (b) R49-D112, and c) R49-E160 calculated from p38α X-ray structures currently deposited in PDB (March, 2017). K53-E71 distances were calculated for structures without any mutations, while R49-D112 and R49-E160 for structures without missing R49, D112, and E160 residues. The average values are shown as red dashed lines, while the bin size is 0.5 Å.

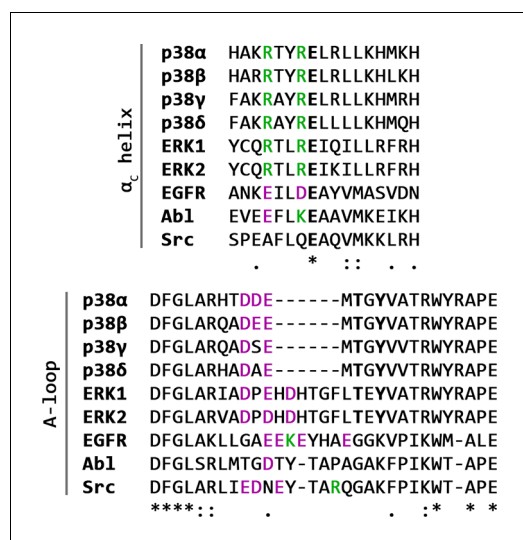

**Figure 10.** Multiple alignments of the α$_C$ helix and the activation loop (from the DFG motif to the P+1 loop) sequences of several human kinases. Selected positively charged residues are colored green, negatively charged ones magenta, while the catalytically important residues in the α$_C$ helix are bolded, as well as the phosphorylated residues in the activation loop.

bridge. ATP binding is therefore only possible in less populated conformations of the protein, explaining the experimentally observed low ATP-binding affinity (*Tokunaga et al., 2014*; *Frantz et al., 1998*). If one looks solely at X-ray structures where R49 is ~15 Å away from D112 (*Figure 9b*), the formation of this particular salt bridge might seem like an artifact arising from force field or methodological imperfections. However, R49 is a part of the loop positioned below the very flexible P-loop (whose residues are often unresolved) and we have often observed its interaction with E160 of the ED site in our simulations (both unbiased and biased), regardless of the protein's phosphorylation state (R49-E160 is ~9 Å in the analyzed X-ray structures - *Figure 9c*). We believe that the reason for these discrepancies lies in the fact that the loop containing R49 forms crystal contacts in all the 206 deposited X-ray structures which would most likely restrict its mobility and prevent the interactions observed in MD simulations.

While the binding of ATP-Mg$^{2+}$ helps p38α come closer to the active form, it is insufficient to lead to a complete shift, as proposed by the NMR model (*Tokunaga et al., 2014*). The full transition, however, does occur in our simulations upon binding of the MK2 docking peptide, which has been previously shown to enhance the catalytic steps (*Tokunaga et al., 2014*). We

hypothesize that the docking peptide prevents the formation of the R49-D112 salt bridge in the phosphorylated protein and brings in the L16 loop. This allows the A-loop to form additional stabilizing contacts, as suggested in the case of ERK2 (*Zhou et al., 2006*) and causes the observed 20-fold increase in the ATP-binding affinity by the MK2 docking peptide (*Tokunaga et al., 2014*). The changes that the docking peptide introduces into the protein structure are particularly interesting when it comes to the crystal contact analysis of the dually phosphorylated p38α X-ray structure with the active A-loop conformation. We found an atypically long expression tag of the neighboring molecule bound in the docking site which, together with other crystal contacts, induces the active state stabilized within the crystal. These observations show how important it is to carefully analyze symmetry-related molecules and they call for caution in the interpretation of deposited X-ray structures, as they can be misleading.

Overall, our extensive PT-metaD calculations converge to free-energy surfaces that together with the unbiased MD simulations offer unparalleled insight into the molecular mechanism of p38α canonical activation elucidating a complex conformational behavior that is consistent with experimental observations. Despite the initial discrepancies between previous structural studies, our findings reconcile the deposited models from X-ray crystallography with the NMR data. Considering the importance of p38α for both normal physiology and pathological processes, we hope the knowledge obtained in this study will help to target the protein with more specificity.

## Materials and methods

### Protein structures and their preparation

The X-ray structure used as the starting conformation for the PT-metaD simulations of inactive unphosphorylated and phosphorylated human p38α is deposited under the Protein Data Bank (PDB) code 3S3I. This structure was also used with the structure of the active phosphorylated mouse p38α (PDB ID: 3PY3) to build the human protein in the latter conformation by using ICM (RRID:SCR_014878) (*Abagyan et al., 1994*). The coordinates of the ATP molecule and the Mg ions were taken from the active phosphorylated human p38γ (PDB ID: 1CM8), while the coordinates of the MK2 docking peptide (residues 370–393) were taken from the p38α-MK2 complex (PDB ID: 2OKR). The missing atoms in 3S3I and MK2 peptide were built with PDB2PQR (*Dolinsky et al., 2004*), while the residues T180 and Y182 were phosphorylated using Vienna-PTM server (*Margreitter et al., 2013*). All proteins consist of the sequence E4-P352 and the standard residue numbering for p38α is applied throughout.

### MD simulations

Each simulated system was generated using CHARMM27 force field (*MacKerell et al., 1998*; *Foloppe and MacKerell, Jr., 2000*; *MacKerell and Banavali, 2000*) at pH 7.4. The protonation states of the residues were determined both by H++ server (*Anandakrishnan et al., 2012*) and PROPKA3.0 (*Li et al., 2005*) which gave identical results leaving all the residues in their usual charge states, except for His312 which was protonated. Since the p$K_{a2}$ of phosphoric acid is measured at 6.7 and 7.2 (*Peacock and Nickless, 1969*; *Lide, 2005*), the charge of phosphoric residues was set at −1. The systems were solvated with ~20,000 TIP3P water molecules (*Jorgensen et al., 1983*; *Mahoney and Jorgensen, 2000*) in a triclinic box with periodic boundary conditions, while Na$^+$ and Cl$^-$ ions were added to reach neutrality and the final concentration of 50 mM (the total number of atoms was ~65,000). Energy of the simulated system was initially minimized in two cycles of steepest-descent energy minimization. The initial velocities for the atoms were taken from Maxwell distribution at 100 K, and the system was subsequently heated to 300 K in five steps of 50 K simulated for 200 ps at constant volume each using velocity rescale thermostat (*Bussi et al., 2007*). In parallel, atomic position restraints for the protein (as well as the ATP molecule and Mg$^{2+}$ ions if present) were uniformly relaxed (with the restraint spring constant going from 25,000 kJ mol$^{-1}$ nm$^{-2}$ to 0 kJ mol$^{-1}$ nm$^{-2}$ in steps of 5000 kJ mol$^{-1}$ nm$^{-2}$). In the end, the system was equilibrated for additional 200 ps under the final conditions. The magnesium ions were restrained to the ATP molecule, that is its last two phosphate atoms (at 3 and 4 Å) and their adjoining oxygen atoms (two restraints per Mg$^{2+}$ at 2.4 Å) throughout the equilibration and production runs using a restraint spring constant of 150 kJ mol$^{-1}$ nm$^{-2}$. The weak restraints were used to make sure that the ions do not unbind at higher temperature replicas. The production

simulations were generated using the GROMACS 4.6.7 biomolecular simulation package (RRID:SCR_014565) (*Pronk et al., 2013*) with the PLUMED2.1 plugin (*Bonomi et al., 2009*; *Tribello et al., 2014*) with a 2-fs integration step. Constant volume conditions were employed throughout with the velocity rescale thermostat (*Bussi et al., 2007*). Bond lengths were constrained using LINCS (*Hess et al., 1997*), while van der Waals interactions were treated with a cutoff of 10 Å. Electrostatic interactions were computed using the particle mesh Ewald method (*Essmann et al., 1995*) with the direct sum cut-off of 10 Å and the Fourier spacing of 1.2 Å.

In the case of unbiased simulations (unphosphorylated p38α with the A-loop in the extended conformation, apo dually phosphorylated p38α with the A-loop in the extended conformation, the dually phosphorylated p38α with bound ATP-Mg$^{2+}$, and the dually phosphorylated p38α with bound ATP-Mg$^{2+}$ and MK2 docking peptide), 10 independent runs of each system were equilibrated to a temperature of 380 K by two additional steps of 40 K compared to the aforementioned protocol, while the position restraints for the protein were uniformly relaxed in steps of 2500 kJ mol$^{-1}$ nm$^{-2}$. The system was equilibrated for additional 200 ps under constant pressure using Berendsen barostat (*Berendsen et al., 1984*). The production simulations were generated under the same conditions as above with the addition of constant pressure of one bar using Parrinello-Rahman barostat (*Parrinello and Rahman, 1981*) and coordinate output of 10 ps. In the case of unphosphorylated system, each independent run was initially 500 ns long. One run that showed a broken $\beta$-sheet motif after 500 ns was extended to a total of 2.675 μs. For the apo phosphorylated system, each independent run was 1 μs long. One of the trajectories showed a broken $\beta$-sheet motif early on and was extended to 2 μs. Each independent run of the phosphorylated system with bound ATP-Mg$^{2+}$ was also initially simulated for 1 μs, while one run that exhibited more conformational diversity was extended to 1.55 μs to transition to the global minimum. In the case of the dually phosphorylated system with bound ATP-Mg$^{2+}$ and MK2 docking peptide, each independent run was 1 μs, amounting to a total sampling time of 10 μs. The magnesium ions were restrained to the ATP molecule throughout the equilibration and production runs in the same way as in the metadynamics runs.

For the simulation of p38α protein crystal, a supercell of the dually phosphorylated mouse p38α (PDB ID: 3PY3) containing four unit cells (2 × 2 × 1), each with four asymmetric units (i.e. 16 protein molecules in total), was built using PyMOL (RRID:SCR_000305) (*Schrodinger, 2015*). Missing residues of the N- and C-terminal tails were added to each protein using Modeller (RRID:SCR_008395) (*Sali and Blundell, 1993*). Protein chain whose tail was built buried within the crystal was chosen as the starting structure to avoid steric clashes caused by the lack of periodic boundary conditions during the modelling process. The supercell was rebuilt in PyMOL and used for pK$_a$ calculations with PROPKA3.0 (*Li et al., 2005*) leaving all the residues in the same charged state as in the previous simulations, while among the newly built residues His-10 was protonated. The system was solvated with ~24,000 TIP3P water molecules (*Jorgensen et al., 1983*; *Mahoney and Jorgensen, 2000*) in a triclinic box using the dimensions of the original crystal and periodic boundary conditions. Na$^+$ and Cl$^-$ ions were added to reach neutrality and the ionic strength of sodium citrate used in the original experiment (with the final concentration of 600 mM). The total number of atoms in the system was ~170,000. The system was minimized and equilibrated as described above to the temperature of 293 K which corresponds to the temperature used for growing crystals. Canonical ensemble at 293 K was maintained for the production run which was 500 ns long. By using a supercell system, crystal contacts were present throughout the simulation, while the interaction of molecules with themselves through periodic boundary conditions was avoided and 8 μs of sampling were obtained for the dually phosphorylated p38α in the crystal environment.

## Enhanced sampling

First, a parallel tempering metadynamics (PT-metaD) with 10 replicas per system was performed at increasing temperatures (300, 310, 320, 330, 341, 352, 363, 375, 387, and 400 K) in which only the potential energy was biased to achieve an exchange rate of 30% between the neighboring replicas. The obtained bias was saved and used further in the production runs as a constant bias. The production runs of PT-metaD in the well-tempered ensemble (*Bonomi and Parrinello, 2010*) were performed for each system consisting of 10 replicas, meaning that a Gaussian potential was deposited in the collective variable space every 2 ps with the height $W = W_0 e^{-V(s,t)/(f-1)T}$, where $W_0$ = 5 kJ mol$^{-1}$ is the initial height, $V(s,t)$ is the bias potential at time $t$ and CV value $s$, $f$ = 10 is the biasing factor, and $T$ is the

temperature of the replica. The collective variables used in the study are distances in contact map space from the inactive (CV1) and active (CV2) A-loop conformation. $CV(R) = 1/N \sum_{\gamma \epsilon \Gamma} (D_\gamma(R) - D_\gamma(R_{ref}))^2$, where $D_\gamma(R)$ is a sigmoidal function that measures the degree of formation of the contact $\gamma$ in the structure $R$ and is defined as $D_\gamma(R) = W_\gamma \frac{1-(r_\gamma/r_\gamma^0)^n}{1-(r_\gamma/r_\gamma^0)^m}$. $r_\gamma$ is the contact distance in the structure $R$, $r_\gamma^0$ is the contact distance in either the reference inactive or active conformation depending on which the contact $\gamma$ is specific for, $W_\gamma$ is the weight of the contact and is set to one for regular contacts and three for salt bridges, $N$ is a normalization constant, n = 6, and m = 10. The widths of the Gaussians are $\sigma_{1,2} = 0.05$. The set of contacts $\Gamma$ defining the contact maps was determined as follows: minimized structures of inactive (PDB ID: 3S3I) and active (PDB ID: 3PY3) conformations were used to determine all the pairs of $C_\alpha$, $C_\beta$ or backbone O atoms that fall within a 5 Å distance cutoff. To relax and lose possible crystal contacts, 10 independent equilibration runs were generated for both the inactive and the active phosphorylated systems. Only those atomic pairs that appear in five or more equilibrated structures and that specifically discriminate between the inactive and active conformations were kept, that is, those that appear either in one or the other, thereby discarding the pairs common to both conformations. A set of 97 regular contacts was obtained this way to which seven salt-bridge contacts involving residues of the A-loop that were specific to the active conformation were added and weighed three times more than regular contacts.

## Convergence of metaD simulations

Initially, FE surfaces were produced with a bias corresponding to an exchange rate of 30% between neighboring replicas and were subsequently refined by lowering the bias on the potential energy to achieve an exchange rate of 10% and run for additional 150 ns to convergence. The simulations were considered to be converged once the free energy in the bidimensional projections and in the monodimensional projections does not change more than 2 kcal mol$^{-1}$ in the last 100 ns. This criterion resulted in a 2.425-µs-long simulation for the unphosphorylated, 2.485-µs-long simulation for the phosphorylated system, 1.035-µs-long simulation for the phosphorylated system with bound ATP, and 0.986-µs-long simulation for the phosphorylated system with bound ATP and MK2 docking peptide.

## Analysis of structures populating the free-energy minima

Free-energy surfaces (FES) for the two CVs were calculated by integrating the bias deposited during the simulations. For each system, several basins from the FES plot were chosen for further analysis. All the structures that fell within each basin were clustered using the *gromos* algorithm (*Daura et al., 1999*) and the *g_cluster* routine (GROMACS) by using the RMSD of $C_\alpha$ atoms as the distance between the structures and the cutoff value of 2 Å. The central structure of the most populated cluster of each basin (i.e. the structure with the smallest distance to all the other members of the cluster) was chosen as the representative of the basin. Furthermore, central structures of top five most populated clusters (which on average contain 75% of all structures) were inspected to make sure that the main structural features in terms of P-loop and A-loop positions were captured by these clusters. To avoid highlighting interactions that occur only sporadically in the simulations, only those salt-bridge interactions that are maintained in the majority of structures of the most populated cluster, as well as most of structures of the selected basin, were highlighted in the tables throughout the paper.

## Analysis of X-ray structures and their symmetry-related neighbors

Symmetry-related molecules of mouse apo p38α-pTpY (PDB ID: 3PY3) that are closer than 4 Å to the original structure were generated by PyMOL (*Schrodinger, 2015*). The same distance criterion was used to determine which pairs of residues of neighboring molecules come into contact.

## Acknowledgements

We thank Dr. R Soliva and Dr. Lucia Diaz Bueno for many helpful discussions. We acknowledge PRACE for awarding us access to resource MareNostrum based in Spain at Barcelona Supercomputing Center. AK was partly supported by the European Union Seventh Framework Programme (FP7/

Marie Curie Actions/COFUND, no. 600404). FLG and GS acknowledge financial support from EPSRC (grant EP/M013898/1). ARN was supported by a grant from the European Commission (ERC, project ID: 294665). MO is an ICREA research fellow. MO acknowledges the support from two grants from the European Commission H2020 program (BioExcel project ID: 676559 and ELIXIR-Excellerate project ID: 675728) and the Spanish MINECO (BIO2015/64802-R). We gratefully acknowledge institutional funding from the Spanish Ministry of Economy, Industry and Competitiveness (MINECO) through the Centers of Excellence Severo Ochoa award, and from the CERCA Programme of the Catalan Government.

## Additional information

### Funding

| Funder | Grant reference number | Author |
|---|---|---|
| Seventh Framework Programme | FP7/Marie Curie Actions/COFUND, no. 600404 | Antonija Kuzmanic |
| Engineering and Physical Sciences Research Council | EP/M013898/1 | Giorgio Saladino Francesco L Gervasio |
| European Research Council | Project ID 294665 | Angel R Nebreda |
| Horizon 2020 | BioExcel (ID: 676559) | Modesto Orozco |
| Horizon 2020 | ELIXIR-Excellerate (ID: 675728) | Modesto Orozco |
| Ministerio de Economía y Competitividad | BIO2015/64802-R | Modesto Orozco |

The funders had no role in study design, data collection and interpretation, or the decision to submit the work for publication.

### Author contributions

AK, Conceptualization, Formal analysis, Validation, Investigation, Visualization, Methodology, Writing—original draft, Writing—review and editing; LS, Conceptualization, Investigation, Methodology; GS, Formal analysis, Investigation, Methodology; ARN, Conceptualization, Supervision, Funding acquisition, Project administration, Writing—review and editing; FLG, Conceptualization, Resources, Supervision, Investigation, Methodology, Writing—review and editing; MO, Conceptualization, Resources, Supervision, Funding acquisition, Project administration, Writing—review and editing

### Author ORCIDs

Antonija Kuzmanic, http://orcid.org/0000-0003-2815-5605
Ludovico Sutto, http://orcid.org/0000-0002-4084-8562
Giorgio Saladino, http://orcid.org/0000-0002-3234-5762
Angel R Nebreda, http://orcid.org/0000-0002-7631-4060
Francesco L Gervasio, http://orcid.org/0000-0003-4831-5039
Modesto Orozco, http://orcid.org/0000-0002-8608-3278

## Additional files

### Supplementary files

• Supplementary file 1. Interactions of key residues in the activation loop observed in the free-energy minima. Residue pairs with interaction occupancy >75% in the most populated clusters of the selected minima are bolded, while the occupancy for the rest of the pairs is in the range of 60–75%.

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
