## [Decision Letter]

Thank you for submitting your article "Molecular mechanism of canonical activation of p38α MAP kinase" for consideration by *eLife*. Your article has been favorably evaluated by John Kuriyan (Senior Editor) and three reviewers, one of whom is a member of our Board of Reviewing Editors.

The reviewers have discussed the reviews with one another and the Reviewing Editor has drafted this decision to help you prepare a revised submission.

Summary:

This manuscript reports simulation studies aiming to elucidate the activation mechanism of p38α kinase. Using unbiased molecular dynamics simulations and calculation of free energy surfaces of the system, the authors attempt to reconcile the apparent contradictions between the X-ray structures of p38α (showing large structural changes associated with kinase dual-phosphorylation) and the NMR findings (the phosphorylation does not lead to significant chemical shift perturbation).

Essential revisions:

1) The free energy surface (FES) of ATP-bound phosphorylated system shows that the active state is accessible now, but unbiased MD simulations showed that only with substrate binding the system is stable at the active basin. To complete the paper's argument, FES of phosphorylated p38α with ATP and substrate bound should be calculated besides the unbiased simulations at 380 K. Also, showing unbiased MD on the phosphorylated and ATP-bound system without substrate and mapping the trajectory to the two CVs analogous to Figure 4 would be helpful to this argument.

2) p38α should be simulated with its crystal contacts in unbiased simulations to support the notion that crystal contacts maintain apo p38α in its otherwise unstable active conformation in crystal environment.

3) The paper is descriptive of selected conformations representative of conformational ensembles at various free energy basins; but, it is not clear how the discussed conformations are selected to represent the ensemble of each basin. The selections should be justified based on characterization of the ensemble of the entire free-energy basin; this justification is missing as is.

Other important points:

1) For phosphorylated p38α, a histogram/time-series should be shown for R49-D112 salt-bridge (ascribed to occlude ATP binding). The regions in the free energy surface allowing ATP binding should also be shown. Reviewers are concerned that the free energy surfaces of the apo unphosphorylated system are very similar to the apo phosphorylated, but the latter system binds ATP much stronger than the former. (In Tokunaga et al., 2014, the ATP affinity is 430 μM, while Frantz et al., 1998 only reported the Km[ATP] to be 25 μM. Unphosphorylated systems binds ATP much weaker at millimolar affinity.) How is the μM ATP binding reconciled with the R49-D112 salt-bridge? Another concern is that, by the energy surface of the apo phosphorylated system, the ATP-bound X-ray structure is as much as 16~18 kcal/mol higher in free energy than the inactive conformation, which seems unlikely.

2) The manuscript reports certain structural features differ considerably from any of the known crystallographic structures (e.g. 3S3I, 3PY3, 2OKR and 1CM8), which calls to question the structural details extracted from the PT-metaD trajectories (e.g. the activation-loop helix (purple in Figure 3, and the R49-D112 salt bridge, which requires D113 to move 18 Å from its positions in the X-ray structures). The reliability of these simulations generated conformations need be calibrated in Discussion.

3) Finally, how the simulation results explain the seemingly contradictory X-ray and NMR data needs to be further discussed in more explicit terms. The contradictory is understood to be the lack of chemical shift perturbation (CSP) between unphosphorylated and phosphorylated yet the x-ray structures are the closed and open alternative structures of the A-loop. The NMR study also shows a large CSP for ATP binding to phosphorylated p38α. The minima in the FE surfaces in Figure 2 differ between the 3 states (unphosphorylated, phosphorylated and ATP bound phosphorylated) yet have overlapping regions. The overlap could explain the lack of CSP for unphosphorylated and phosphorylated (as stated in the subsection “Dually phosphorylated p38α”), but how the large CSP for bound ATP is rationalized by the FE surface, since, as stated in the manuscript, "the representative structure of the global minimum [for ATP-bound] is still not too different from the dominant one in the apo phosphorylated state." This should be explained in more explicit terms.

---

## [Author Response]

*Essential revisions:*

*1) The free energy surface (FES) of ATP-bound phosphorylated system shows that the active state is accessible now, but unbiased MD simulations showed that only with substrate binding the system is stable at the active basin. To complete the paper's argument, FES of phosphorylated p38α with ATP and substrate bound should be calculated besides the unbiased simulations at 380 K. Also, showing unbiased MD on the phosphorylated and ATP-bound system without substrate and mapping the trajectory to the two CVs analogous to Figure 4 would be helpful to this argument.*

Following reviewers’ suggestions, we have now complemented our manuscript with the FES of the phosphorylated p38α with bound ATP and the MK2 docking peptide (with overall sampling of >9 µs). We have changed Figure 2 and its supplements accordingly, as well as the manuscript text to reflect the addition of new data. We have also performed unbiased MD simulations of the phosphorylated p38α with and without bound ATP which resulted in two new figures (Figure 4 and Figure 5) and >20 µs of sampling of these systems. What is more, we have extended our unbiased simulations of the phosphorylated system with bound ATP and the MK2 docking peptide (reaching 10 µs of sampling) whose results are shown in an updated version of Figure 4 (Figure 6 in the revised version). The additional simulations give further support to the claims we have made in the original submission.

*2) p38α should be simulated with its crystal contacts in unbiased simulations to support the notion that crystal contacts maintain apo p38α in its otherwise unstable active conformation in crystal environment.*

We thank the reviewers and the editor for the useful advice. The results of the new simulations strengthen the overall conclusions. We have simulated a crystal supercell of the dually phosphorylated mouse p38α (PDB ID: 3PY3) containing 4 unit cells (2x2x1), each with 4 asymmetric units (16 copies of the protein in total). By using a supercell system, we have been able to maintain crystal contacts, avoid the interaction of molecules with themselves through periodic boundary conditions, and achieve 8 µs of sampling (500 ns of the whole supercell x 16 copies of the protein). We have described the modelling process and the simulation setup in the Methods section (subsection “MD simulations”, last two paragraphs), added the results as Figure 8, and made appropriate changes in the text referring to new data (subsection “The active conformation”, fourth paragraph).

As shown by Figure 8, the A-loop is very stable in the conformation captured by the X-ray model – the β-sheet motif is unperturbed during the simulation, as well as the contacts made by pT180 with the three conserved arginines (R67, R70, and R173). However, pY182 is more mobile and at times breaks away from H228 and the conserved arginines (R186 and R189). This is consistent with the electron density which is not so well defined for pY182 as it is for the rest of the A-loop (see Figure 11). Finally, we show an excellent qualitative agreement with the B-factors reported for the X-ray structure (3PY3), while (as expected) the most mobile regions in our simulations (mainly, the N- and C-terminal tails) are missing from the deposited electron density. While comparing the crystallographic and MD-derived B-factors it is also important to keep in mind that: 1) The data were collected at 100 K. Since MD force-fields and algorithms have not been developed for such low temperatures, we have performed our crystal simulations at 293 K which proportionally increases the overall atomic motions; 2) The experimental data collected for the crystals in general are time and ensemble averages of 10^23^ molecules which are in the end represented by a single model, while its B-factors then reflect both the static and the dynamic disorder of the crystal. Indeed, two independent studies had shown that the refinement process underestimates the disorder present in the crystal thereby producing lower B-factors (Janowski et al., 2013, JACS, doi: 10.1021/ja401382y; Kuzmanic et al., 2014, Nat. Commun., doi: 10.1038/ncomms4220).

Author response image 1.**DOI:**
http://dx.doi.org/10.7554/eLife.22175.021

*3) The paper is descriptive of selected conformations representative of conformational ensembles at various free energy basins; but, it is not clear how the discussed conformations are selected to represent the ensemble of each basin. The selections should be justified based on characterization of the ensemble of the entire free-energy basin; this justification is missing as is.*

We analyzed each selected free energy basin through a clustering method (as described in the Methods section) and we chose the central structures of the most populated clusters as the representative ones we had shown and discussed in the paper. Furthermore, we checked the central structures of top 5 most populated clusters (which on average contain 75% of all structures) to make sure that the main structural features in terms of P-loop and A-loop positions were captured by these clusters. To avoid highlighting interactions that occur only sporadically in the simulations, we also added the tables (Table 1 and [Supplementary-material SD1-data]) with salt-bridge interactions that are maintained in the majority of structures of the most populated cluster, as well as most of structures of the selected basin. In order to avoid confusion among the readers, we now state clearly in the main text how the structures were chosen (Results, first paragraph and subsection “Enhanced sampling”). We have also modified the aforementioned tables that now include a measure of how often the interaction occurs in the most populated cluster of each bin.

*Other important points:*

*1) For phosphorylated p38α, a histogram/time-series should be shown for R49-D112 salt-bridge (ascribed to occlude ATP binding). The regions in the free energy surface allowing ATP binding should also be shown. Reviewers are concerned that the free energy surfaces of the apo unphosphorylated system are very similar to the apo phosphorylated, but the latter system binds ATP much stronger than the former. (In Tokunaga et al., 2014, the ATP affinity is 430 μM, while Frantz et al., 1998 only reported the Km[ATP] to be 25 μM. Unphosphorylated systems binds ATP much weaker at millimolar affinity.) How is the μM ATP binding reconciled with the R49-D112 salt-bridge? Another concern is that, by the energy surface of the apo phosphorylated system, the ATP-bound X-ray structure is as much as 16~18 kcal/mol higher in free energy than the inactive conformation, which seems unlikely.*

Following reviewers’ suggestions, we have now included the histogram for the R49-D112 salt bridge as Figure 2—figure supplement 4, which shows that almost all the structures in the most populated cluster of the global minimum have this interaction formed (>16000 structures). We have also highlighted the minimum in the free energy surface of the phosphorylated p38α which is represented by structures that we believe can bind ATP (and which are also explored by the unbiased simulations).

Furthermore, as reviewers have noted, the free energy surfaces of apo unphosphorylated and apo phosphorylated p38α are indeed similar, but they are not the same, nor are their minima populated by the same conformations. In general, free energy surface of a protein reflects its ability to populate certain states defined by collective variables. This means that at a given temperature all the conformations of the system that we have highlighted in the paper will be accessible; however, their population will be different.

Based on our analyses, apo unphosphorylated p38α does not seem to explore conformations that can bind ATP – either the highly flexible P-loop is collapsed into the ATP-binding site, or the R173 of the A-loop forms interactions that are typically viewed as autoinhibitory (e.g., forming a salt bridge with D168 of the DFG motif which stabilizes ATP upon binding). Upon the A-loop phosphorylation, two residues change their properties significantly which allows them to stabilize interactions which are inaccessible to the unphosphorylated system. However, the global minimum of the phosphorylated system is dominated by the conformation with the formed R49-D112 salt bridge which would prevent ATP binding. Nevertheless, this does not mean that the phosphorylated p38α cannot bind ATP, seeing how a higher-energy minimum is populated by conformations that can bind ATP (which are also explored by the unbiased simulations). Basically, once p38α is phosphorylated, it exists as a mixture of all the described conformations that are present at different concentrations, therefore determining the macroscopic property of ATP binding to the phosphorylated system. This agrees with the experimental observations mentioned by the reviewers, as well as the observation that the binding of the MK2 docking peptide has a profound effect on ATP binding, i.e. it causes a 20-fold increase in binding which most likely occurs due to the breaking of the salt bridge and the overall stabilization of the α_D_ helix. However, our results do not exclude the possibility of induced fit mechanism in which the addition of ATP to the phosphorylated system would shift the populations towards structures that are more likely to bind ATP.

Finally, we have shown with our metadynamics simulations (and confirmed additionally with unbiased MD simulations) that the apo phosphorylated structure captured by X-ray crystallography is 16-18 kcal/mol higher in energy and therefore rather inaccessible to the phosphorylated system in the absence of ATP. However, the presence of bound ATP in the system causes a shift towards this conformation which is then at 2-4 kcal/mol compared to the global minimum, not at 16-18 kcal/mol. This observation agrees with the NMR data which show large chemical shift perturbations only upon ATP binding, not the dual phosphorylation.

We have amended our manuscript throughout to discuss the points above more clearly (e.g., subsections “Unphosphorylated p38α” and “Dually phosphorylated p38α with bound ATP-Mg^2+^”, and Discussion).

*2) The manuscript reports certain structural features differ considerably from any of the known crystallographic structures (e.g. 3S3I, 3PY3, 2OKR and 1CM8), which calls to question the structural details extracted from the PT-metaD trajectories (e.g. the activation-loop helix (purple in Figure 3, and the R49-D112 salt bridge, which requires D113 to move 18* Å *from its positions in the X-ray structures). The reliability of these simulations generated conformations need be calibrated in Discussion.*

We agree with the reviewers that there is always a possibility that certain structural features arise from either force field or methodological imperfections, we have thus modified the Discussion which now mentions these possible pitfalls (Discussion, first and second paragraphs). However, we also find that crystal contacts introduce artifacts that can lead to misinterpretation of the structural features of p38α (as we have already described above for 3PY3). When it comes to the position of the A-loop in the unphosphorylated state, out of 206 currently deposited X-ray structures of human p38α (with 224 individual chains), only 47 molecules have complete A-loops (i.e., without any missing or zero occupancy residues). From this reduced set, a single deposited structure has the A-loop in a conformation in which the loop does not form any crystal contacts (PDB ID: 2BAJ). Unfortunately, this structure has been solved in the DFG-out conformation with a bound inhibitor and it cannot be representative of the apo unphosphorylated state. However, the mere lack of information indicates just how flexible the A-loop is and why it is almost impossible to resolve it unless it forms crystal contacts. Thus, it is not surprising that the A-loop conformations we observe do not match the ones present in X-ray structures. Furthermore, we have shown that extensive crystal contacts are in fact responsible for the conformation observed in 3PY3 (the only X-ray structure of the dually phosphorylated p38α), while we suspect that the same applies for 1CM8 (dually phosphorylated p38γ with bound ATP). While this structure does not have anything bound in the docking site, its A-loop forms crystal contacts with the N-terminal loops of a symmetry related molecule (see Figure 12). Unfortunately, the electron density of this molecule is unavailable, so we are unable to inspect the quality of the modelled A-loop and its crystal contacts. We have also included this analysis in the manuscript (Discussion).

Author response image 2.Contact maps for interactions formed between the residues of the reported X-ray structure (PDB ID: 1CM8, p38γ-pTpY) (x-axis) and its symmetry related neighbors (y-axis).The residues are considered to be in contact if the minimal distance between any of their atoms is under 4 Å. The contacts formed by the A-loop are indicated by the solid ellipse. Secondary structure elements are also shown with α-helices in purple, β-sheets in green, and loops in grey. Missing residues are shown as light grey rectangles.**DOI:**
http://dx.doi.org/10.7554/eLife.22175.022

As for the position of R49 and its interactions, the deposited X-ray structures are again quite misleading. R49 is a part of the loop positioned below the very flexible P-loop (whose residues are often unresolved) and is on average 9 Å away from the C-lobe residues in the deposited X-ray structures. However, in our simulations (both unbiased and biased), we have often observed R49 interacting with E160 of the ED site, regardless of the protein’s phosphorylation state. We believe that the reason for this discrepancy (as well as the formation of the R49-D112 salt bridge) lies in the fact that R49 loop forms crystal contacts in all 206 deposited X-ray structures which would most likely restrict its mobility and prevent the interactions observed in MD simulations. We have included these findings in the revised version of the manuscript (Discussion, second paragraph).

*3) Finally, how the simulation results explain the seemingly contradictory X-ray and NMR data needs to be further discussed in more explicit terms. The contradictory is understood to be the lack of chemical shift perturbation (CSP) between unphosphorylated and phosphorylated yet the x-ray structures are the closed and open alternative structures of the A-loop. The NMR study also shows a large CSP for ATP binding to phosphorylated p38α. The minima in the FE surfaces in Figure 2 differ between the 3 states (unphosphorylated, phosphorylated and ATP bound phosphorylated) yet have overlapping regions. The overlap could explain the lack of CSP for unphosphorylated and phosphorylated (as stated in the subsection “Dually phosphorylated p38α”), but how the large CSP for bound ATP is rationalized by the FE surface, since, as stated in the manuscript, "the representative structure of the global minimum [for ATP-bound] is still not too different from the dominant one in the apo phosphorylated state." This should be explained in more explicit terms.*

As also discussed above, the free energy surfaces do not only reflect which areas have been explored by p38α in MD simulations, but also how often. This means that even though the free energy surface of the phosphorylated system with bound ATP overlaps with the apo one, these systems predominantly explore different conformations. Upon ATP binding the A-loop can finally explore regions that are almost inaccessible in the absence of ATP and it also forms more stable contacts with the α_C_ helix in the global minimum. These changes, along with the fact that ATP is in direct contact with the binding site residues, would explain the differences observed by NMR. We have modified the manuscript text to clarify these points (e.g., subsection “The active conformation”).